# Preparation and Characterization of Cellulose Nanofibers from Banana Pseudostem by Acid Hydrolysis: Physico-Chemical and Thermal Properties

**DOI:** 10.3390/membranes12050451

**Published:** 2022-04-22

**Authors:** Mohammad Sobri Merais, Nozieana Khairuddin, Mohd Harfiz Salehudin, Md. Bazlul Mobin Siddique, Philip Lepun, Wong Sie Chuong

**Affiliations:** 1Department of Science and Technology, Faculty of Humanities, Management and Science, Universiti Putra Malaysia Bintulu Sarawak Campus, Bintulu 97008, Malaysia; sobrimerais@gmail.com (M.S.M.); wongsie@upm.edu.my (W.S.C.); 2Institut Ekosains Borneo, Universiti Putra Malaysia Bintulu Sarawak Campus, Bintulu 97008, Malaysia; 3Department of Bioprocess and Polymer Engineering, School of Chemical and Energy Engineering, Faculty of Engineering, Universiti Teknologi Malaysia, Skudai 81310, Malaysia; mharfizutm@gmail.com; 4Faculty of Engineering, Computing and Science, Swinburne University of Technology Kuching, Kuching 93050, Malaysia; msiddique@swinburne.edu.my; 5Department of Forestry Science, Faculty of Agricultural Science and Forestry, Universiti Putra Malaysia Bintulu Sarawak Campus, Bintulu 97000, Malaysia; philip@upm.edu.my

**Keywords:** bioplastic, biodegradable film, banana pseudostem, nanofiber, acid hydrolysis

## Abstract

Cellulose is a biopolymer that may be derived from a variety of agricultural wastes such as rice husks, wheat straw, banana, and so on. Cellulose fibril that is reduced in size, often known as nanocellulose (NC), is a bio-based polymer with nanometer-scale widths with a variety of unique properties. The use of NC as a reinforcing material for nanocomposites has become a popular research issue. This research paper focuses on the production of banana pseudostem cellulose nanofiber. Nano-sized fiber was obtained from banana pseudostem through several processes, namely, grinding, sieving, pre-treatment, bleaching, and acid hydrolysis. The product yield was found to be 40.5% and 21.8% for *Musa acuminata* and *Musa balbisiana*, respectively, by the weight of the raw fiber. The reduction in weight was due to the removal of hemicellulose and lignin during processing. Transmission electron microscopy (TEM) analysis showed that the average fiber size decreased from 180 µm to 80.3 ± 21.3 nm. Finally, FTIR analysis showed that the fibers experienced chemical changes after the treatment processes.

## 1. Introduction

Biodegradable polymers made from renewable resources have received a lot of interest as next-generation packaging materials. Because of their biodegradability and renewability, polysaccharides have the potential to replace petrochemical polymers. Cellulose is the most abundant biomass on the planet. Micro-fibrillated celluloses have been fragmented into micro/nano-sized fibers and used to make transparent packaging membranes [1]. Nanocellulose (NC) has various benefits over micro- and macro-cellulose composites. For example, NCs are cheap, biodegradable, and renewable with high surface reactivity as well as low density and low energy consumption. Likewise, NCs improve various properties when introduced as reinforcements in nanocomposites. NC composites are widely employed in the automotive, packaging, electronics, and biotechnology sectors, among others, due to their tolerability, style flexibility, and processability. However, there are significant drawbacks to using NCs as a reinforcing material including excessive wetness absorption, poor wettability, incompatibility with most chemical compound matrices, and process temperature limits [2]. These drawbacks have prompted researchers to address these issues through the introduction of numerous processing methods. For example, the NC or compound matrices have been developed or modified using entirely new process techniques, which could supply superior NC-reinforced composites with fascinating properties [2]. Plant cellulose has been used as a natural reinforcer in food packaging films because it is a natural polysaccharide polymer with good thermal stability, and numerous recent studies have shown that incorporating plant-derived celluloses into protein- or starch-based films can improve the mechanical properties [3]. In this study, a biofilm reinforced with NC was extracted from the inner and outer layers of the banana pseudostem.

In the banana plant, the pseudostem is responsible for supplying and transporting nutrients from the soil to the fruits. However, the pseudostem becomes waste biomass after the banana fruit has grown and been processed, which makes the plant unusable for the following harvest [4]. Furthermore, 10% of each tonne of banana fruit collected is discarded resulting in approximately four tonnes of biomass waste. The root, pseudostem, rotting fruit, peel, fruit-stem, and rhizome are all sources of biomass waste. This indicates that each phase of banana fruit production results in four times the amount of biomass waste [5]. Banana farms yield 220 tonnes of biomass waste per hectare [6]. The leftovers are considered garbage and must be handled properly, otherwise these can generate greenhouse gases if burnt or abandoned, thereby causing environmental issues [7]. However, crop wastes can be utilized for cellulose fibers [8].

Of all the banana species in the world, *Abaca* (*Musa textilis*) is the most widely recognized for its leaf-based fiber. Meanwhile, *Musa acuminata* is the most consumed banana [7]. The pseudostem of banana species has high tensile strength and rigidity, which indicates it is a promising fiber material [9]. The use of CNFs as matrix support components improves the thermo-mechanical characteristics, reduces polymer exposure to water, and maintains biodegradability [10]. The mixing of CNFs with polysaccharides (such as starch) improves the general matrix characteristics [11,12,13]. However, the extraction of CNFs from lignocellulosic sources requires several methods. Currently, the isolation of nanofibril cells is accomplished through chemical hydrolysis (e.g., acid therapy and catalytic oxidation), which separates cellulose fibers from the plant cell wall [14]. Other ways to obtain CNF involve high-intensity ultrasonication [15]. The primary goal of this research is to acquire nano-sized fibers from banana pseudostem through an acid hydrolysis process. Secondly, the structure of banana pseudostem-based CNFs will be investigated. Finally, the chemical and thermal characteristics of banana pseudostem CNF will be examined.

## 2. Materials and Methods

### 2.1. Grinding and Sieving

The isolation of CNFs from banana pseudostem requires several steps. First, the partially-dried banana trunks were obtained from Universiti Putra Malaysia Kampus Bintulu, Sarawak. Upon receipt, the banana trunks were dried in a pressure convection oven at 60 °C for 48 h. Figure 1 shows the differences between the dried *Musa acuminata* and *Musa balbisiana*. The dried *M. balbisiana* pseudostem appears darker when compared to *M. acuminata*, which may indicate higher lignin and cellulose content. A previous study indicates that the black-colored substances obtained from the pulping process is mostly due to lignin (46% of the total solid) and hemicellulose [16]. Another study also reported that the wood fiber from *M. balbisiana* has significantly higher lignin and hemicellulose contents [17].

The banana pseudostem was ground in a household blender for 30 s with ten-second pauses in between. The ground banana pseudostem was then sieved using laboratory test sieve ASTM E11. Several sieves were arranged accordingly from larger to smaller mesh sizes; 1 mm, 500 µm, 355 µm, and 180 µm, and a collector. The sieves were put on a shaker for 2 min. Figure 2 shows the weight distribution according to each mesh size.

### 2.2. Pre-Treatment

The pre-treatment and bleaching processes eliminate hemicellulose, lignin, and other substances such as wax, which purifies the extracted cellulose fibers [18]. Next, 20 g of fiber with a size <180 µm was immersed in hot distilled water (*d*H_2_O), also called the retting process, at temperatures between 70 °C and 80 °C in a water bath. The cycle was completed in 12 h. This step was done to eliminate impurities and large particles including foreign materials [19]. The fibers were then washed with water and filtered using cotton mesh (soy filter mesh). It was then dried in the oven for 24 h at 60 °C. The fibers were then sterilized by autoclaving at 121 °C for 15 min and treated with 200 mL of 2.5 mol L^−1^ sodium hydroxide (NaOH). This approach, known as the steam explosion method, was developed to remove hemicelluloses [20]. At this point, filtration using filter paper or cotton mesh did not work as it caused clogging. Hence, the centrifugal technique was introduced to replace the previously reported separation method [21]. The NaOH solution was removed from the fiber by centrifuging for 15 min at 10,000 RPM before washing with water. The fiber was rewashed with distilled water after the water residue was removed. The treatment was carried out at least five times or until the pH level reached 7. The water was then evacuated before moving on to the next step.

### 2.3. Bleaching Process

After pre-treatment, the fiber was subjected to bleaching. This was accomplished by soaking 200 mL of sodium chlorite in NaClO_2_ solution at 70 °C under acidic conditions (pH 4–5) [22]. The acidic condition was adjusted by adding 2 mL of glacial acetic acid. The fiber became whiter due to the elimination of lignin and trace hemicellulose. The residue, on the other hand, turned yellowish in color. The residue of the bleaching chemical was centrifuged out of the fiber, and the fiber was well washed with distilled water. This technique was carried out again (5–7 times) until the pH of the treated fiber reached 7 [23]. Before the acid hydrolysis procedure began, the treated fiber was dried at 37 °C and stored.

### 2.4. Acid Hydrolysis Process

CNFs were isolated in order to produce nano-sized cellulose fibers, which can be obtained by hydrolysis and concentrated acid treatment [22]. Firstly, 64 wt% of 105 mL of H_2_SO_4_ solution was added with 10 g of treated cellulose fiber. The mixture was rapidly agitated at 45 °C for 90 min. The hydrolysis process was then terminated by the addition of 400 mL of chilled water. Later, the precipitate was obtained by diluting the suspension and centrifuging at 11,000 RPM for 10 min. The mixture was resuspended in water; this time with a lot of agitation. It was then placed in a Supelco dialysis tube and immersed in water for 7 days to remove any remaining acid. The water was changed every day. Finally, the CNF suspension was homogenized for 30 s using the homogenizer (Model: IKA Ultra Turra T-25, Staufen, Germany). The CNF was stored in the refrigerator prior to further analysis.

### 2.5. Composition of Hemicellulose, Cellulose, Lignin, and Extractives of Banana Pseudostem Fibre after Pre-Treatment, Bleaching and Acid Hydrolysis

The composition of three lignocellulosic components (hemicellulose, cellulose, and lignin) in the banana pseudostems was determined by the method proposed in the literature [24,25].

#### 2.5.1. Determination of Extractives Content

In this step, 60 mL acetone was added to 1 g of banana pseudostem fiber to determine the composition of extractives in the biomass sample. The temperature was kept at 90 °C for 2 h using a heater. The sample was then dried in an oven at 105–110 °C for 2 h until it reached a consistent weight (B). The composition of extractives was computed using the formula below.
A − B = Extractives content (g)(1)

#### 2.5.2. Determination of Hemicellulose Content

In this step, 0.5 mol/L of NaOH solution was added to 1 g of the extractive-free banana pseudostem sample to assess the hemicellulose content in the biomass sample (B). The mixture was refluxed at 80 °C for 310 min. The sample was then rinsed in *d*H_2_O until it was clear of Na^+^. Using pH paper, the Na^+^ was detected and the measurement was reduced to 7. The sample was dried in an oven at 105–110 °C until it reached a consistent weight (C). The hemicellulose content was calculated using the formula below.
(B − C) = Hemicellulose content (g)(2)

#### 2.5.3. Determination of Lignin Content

Next, 1 g of the extractive-free banana pseudostem was treated with 30 mL of 98% H_2_SO_4_ (B). The sample was kept at room temperature for 24 h. It was then heated for 1 h at 100 °C. The solid residue was filtered and rinsed with *d*H_2_O until the sulphate ion was undetectable. The sulphate ion was detected by a titration procedure using a 10% solution of BaCl_2_. The sample was dried in an oven at 105–110 °C until it reached a consistent weight (D). The lignin content was calculated from the ultimate weight of the residue.
(D) = Amount of Lignin (g)(3)

#### 2.5.4. Determination of Cellulose Content

The total quantity of biomass sample utilized in the experiment was 1 g. The cellulose (E) content was determined by calculating the difference between the sample’s original weight and the three other component weights measured during the experimental phase.
(A − B) + (B − C) + D + E = 1 g(4)

### 2.6. Analysis of Banana Pseudostem Cellulose Nanofibers

#### 2.6.1. Thermogravimetric Analysis

Thermogravimetric analysis tests were performed in a TGA 4000 Thermogravimetric Analyzer, Perkin Elmer, Waltham, MA, USA and analyzed with Pyris™ software (Version 11.1.1.0492, Shelton, CT, USA). Approximately 5 mg specimens were weighed in aluminum pans and heated from 30 °C to 900 °C at a heating rate of 10 °C/min with nitrogen purging. The weight losses of the samples were measured as a function of temperature. Nitrogen was purged at a flow rate of 120 mL/min to provide an inert atmosphere for non-isothermal thermal degradation and to remove gaseous and condensable products, thus minimizing any secondary vapor-phase interactions.

#### 2.6.2. X-ray Diffraction

The crystallinity index of both species’ banana pseudostem CNFs was determined using X-ray diffraction (Rigaku SmartLab Intelligent X-ray Diffraction System, Tokyo, Japan) under the following conditions: monochromatic Cu K radiation of 15.418 nm; 40 kV, 40 mA with a step size of 0.02° and time/step around 20 s from 3° to 100° (2). An empirical technique was used to calculate the crystallinity index (CrI) using the following equation [26]:(5)$$\;\mathrm{CrI}\;\;=\;\frac{\left(I_{002}\;-\;I_{\mathrm{am}}\right)}{I_{002}}\;\times \;100$$
where I_002_ is the maximum intensity of the (002) lattice diffraction at 2θ = 22.5° and I_am_ is the intensity of the scattered diffraction by the amorphous part of the sample at 2θ = 18°.

#### 2.6.3. Macroscopic Observation

First, the macroscopic observation of the fiber morphology was recorded using an Olympus SZ51-RT (Olympus, Tokyo, Japan) stereo zoom microscope, attached with an industrial microscope camera ToupTek S3CMOS, with a 5-megapixel sensor, 2.2 × 2.2 µm. The image was captured at 250 magnification of the original specimen size with an auto white balance and exposure setting. The captured images were analyzed using the image analysis software ToupView version 3.7. Secondly, the CNF morphology was also determined using high-resolution transmission electron microscopy (TEM) (Hitachi H-7700 Philips; 120 kV) (HRTEM). A drop of the liquid nanofiber suspension was added to the carbon-coated grid and left to dry at room temperature. The scale measurements of the cellulose nanofiber were carried out using the Image J scanner before and after the homogenization process. In total, ten nanofibers were randomly picked, measured and the result was stated as the mean value of the sample. Lastly, the chemical composition of the cellulose nanofiber was determined using Fourier transform infrared spectroscopy (FTIR) analysis. Dried CNF was dried overnight at 50–60 °C for 120 min before analysis. KBr pellet was used in sample preparation for FTIR analysis. The spectra recorder (FTIR Nicolet MAGNA-IR 860 spectrometer; Thermo Fisher Scientific, Inc.; Waltham, MA, USA) with absorbance mode, resolution: 4 cm^−1^, 64 scans were utilized to record the spectra.

## 3. Results and Discussion

### 3.1. Yield after Pre-Treatment and Acid Hydrolysis

The product yield after each process were recorded gravimetrically using a weighing balance. Based on the results, the yield of CNF obtained was 40.5% (*M. acuminata*) and 21.8% (*M. balbisiana*) per weight of raw fiber. Each CNF extraction process removed a specific composition of holocellulose, lignin, and ashes in the fiber [27]. Likewise, losses occurred during transferring, sieving, and washing throughout the treatments and processes as well as during the estimation of hemicellulose and lignin removal (in weight). Table 1 shows the CNF yields after each process for both *M. acuminata* and *M. balbisiana*.

### 3.2. Morphology

#### 3.2.1. Banana Pseudostem Fibers after Pre-Treatment

The initial diameter of the raw fiber was 355 µm (on average). The observations revealed that the physical improvements occurred in banana pseudo stem fiber due to the significant removal of hemicellulose and lignin fractions during the pre-treatment. After alkali treatment (with steam explosion), the fiber structure started to “unwind”. The fiber was distorted and started breaking down. Therefore, the pre-treatment process removes substantial portions of hemicellulose and lignin fiber materials, leaving cellulose unbound and discarded. A smaller strand started to appear. In comparison, the untreated fiber, still has a pecking structure (rigid form) with highly ordered fibrils since hemicellulose and lignin still bind the microfibrils together. The bleaching process caused the structure to further unwind, making the diameter of the cellulose strand smaller. Further removal of lignin also caused the fiber to become whiter. The images of the CNFs for each process are shown in Figure 3 below.

#### 3.2.2. Transmission Electron Micrograph (TEM) Analysis of *Musa acuminata* CNF

TEM was utilized to verify the nano structure of the fiber. Figure 4 and Figure 5 show that CNFs undergoing acid hydrolysis have thinner fiber strands with a 28–87 nm diameter compared to the raw sample before the process. The amorphous area of microfibril (cleaved during hydrolysis with sulfuric acid), decreased the micrometer-sized fiber to nanometers [28,29,30]. The homogenization turned the fiber into NC with a web-like shape.

#### 3.2.3. Comparison of *M. acuminata* and *M. balbisiana* Pseudostem via XRD Analysis

The degree of crystallinity of the CNF fashioned from *M. acuminata* and *M. balbisiana* pseudostem was delineated by the intense peaks detected at 16° and 22° in the diffractogram shown in Figure 5. The results display the narrowest/sharpest peaks at 2θ = 22° due to high crystallinity compared to other samples. The expansion and realignment of monocrystals may occur simultaneously throughout the CNC generation, which enhances the polyose crystallinity [31]. Figure 6 shows that the crystallinity index of CNF obtained from *M. acuminata* is 65.68%, whereas *M. balbisiana* is higher at 75.37%. It shows that the CNF obtained from *M. balbisiana* has a higher crystallinity region than *M. acuminata*. The removal of hemicellulose, lignin, and amorphous non-cellulosic substances during the pre-treatment and bleaching processes increases the crystallinity due to bonding interactions and Van der Waals forces between close molecules and crystalline structure of saccharide [28]. In comparison with other experiments, 64.32% crystallinity was attained in a comparable experiment using banana rachis. Meanwhile, from raw to treated fiber, crystallinity increases by 78% for banana rachis and 450% for banana peel bran. However, the crystallinity of generated nanofibers from poplar wood was found to be 69% [32]. In addition, softwood pulp has a 78.3% crystallinity index while hardwood pulp has an index of 72.7% [33]. As a result, softwood pulp is projected to have greater cellulose purity than *M. balbisiana*.

The acid hydrolysis approach is the most well-known and commonly utilized of the several methods for generating cellulose nanostructures. This technique separates the cellulose’s disordered and amorphous portions, yielding single, well-defined crystals. The fact that the crystalline areas are insoluble in acids in the conditions under which they are used supports this event [34]. Because the structure of ribbon-shaped crystalline bundles in cellulose is not easily pierced by acid molecules, too brief hydrolysis processes will not result in substantial changes in the crystallinity of the material. On the other side, if the reaction period is too lengthy, the crystalline domains of BC will be digested, resulting in a loss in crystallinity. Previous research has shown that in order to create a material with a crystallinity index >80%, longer hydrolysis periods than those employed for plant cellulose are necessary [35].

### 3.3. Fourier Transform Infrared Spectroscopy (FTIR) of Banana Pseudostem CNF after Pre-Treatment and Processing

The FTIR spectra showed that structural modifications appeared after the sequential fiber treatments. The comparison of selected bands for each fiber (untreated, after alkali treatment, bleached and CNF) are shown in Figure 7 and Figure 8.

For *M. acuminata*, the most significant contrast is the lack of vibration at 1238 cm^−1^. The low value at 1242 cm^−1^ is associated with the C-O-C extension of the lignin aryl-alkyl ether bond [26]. The two peaks disappear from the spectrum of bleached pulp fibers and nanofibers. This was caused by the removal of lignin during chemical treatment. The peak at 1154 cm^−1^, on the other hand, is because of C-O-C vibration hemicellulose [36]. The peak slowly disappeared as the treatment process proceeded. The disappearance of the peak in the graph indicates that a large portion of hemicelluloses was removed during the pre-treatment and bleaching process. The peak, however, disappeared after the hydrolysis process. This suggests the acid hydrolysis process continuously removed hemicelluloses and lignin from the fiber. The peak 1154 cm^−1^ also indicates the vibration band for an amorphous substance [37]. Thus, it also suggests that the acid hydrolyzing process produces highly crystalline cellulose and reduces the amorphous content. In addition, the 1640 cm^−1^ peak in the spectrum of nanofibers and bleached pulp fibers is associated with cellulosic water absorption [38]. As for *M. balbisiana*, the broad region 3700–3200 cm^−1^ is related to OH vibrations. The solid and broad band at 3413 cm^−1^ was attributed to the stretching vibration of hydroxyl groups (-OH).

### 3.4. Thermogravimetric Analysis (TGA) of Commercial and Native Banana Pseudostem

The TGA curves for the *M. acuminata* and *M. balbisiana* pseudostem CNFs are shown in Figure 8. Only one degradation step was observed for both materials. The CNF from both sources generally started its decomposition easily, while the weight loss mainly happened at 263–365 °C. From the observation, the weight loss for both samples are related to evaporation and evacuation of absorbed and bound water [39]. The CNF from *M. acuminata* pseudostem started to degrade at 263 °C, whereas the *M. balbisiana* pseudostem started to degrade at higher temperature 280 °C indicating higher thermal stability. The degradation temperature difference is caused by the higher crystallinity of the CNF, which resulted in the further elimination of hemicelluloses and pectin. However, the difference in the solid residue from both samples is not statistically significant (~2.5–2.7% solid residue) for both. The pyrolysis of non-cellulosic materials produces a higher residual weight [40]. Thus, the low residual weight of CNF from both pseudostems indicate that the sample is highly cellulosic.

The initial decomposition temperature of CNF with some modifications, such as hexagonal boron nitride (h-BN), which is an excellent thermally conductive and electrically insulative material, is 205 °C, which demonstrates acceptable thermal stability and mechanical properties for electronics as a thermal interface and packing material [41]. In addition, modified aerogels containing methylene diphenyl dissocyanate (MDI) reached 353.6 °C. Because of the low oxygen concentration of the aromatic moiety in the MDI structure, it is thought that the mass was preserved in modified lignocellulosic nanofibril (LCNF) aerogel at the same temperature. Additionally, since there were more MDI structures, the modified aerogel had greater residual mass after pyrolysis with a higher MDI content. These comparisons clearly demonstrate the benefits of using MDI to maintain CNF’s thermal stability [42].

### 3.5. Materials Composition of Banana Stem Fiber before Pre-Treatment and Bleaching

The results depicted in Table 2 show that the polyose fraction in the treated fiber was enhanced to 93.16% for *M. acuminata* and 93.36% for *M. balbisiana*, whereas the hemicellulose, polymer and extractives contents was reduced. This high cellulose content provides toughness, strength, rigidity, and structural constancy to the banana pseudostem fiber, whereas hemicellulose decreases the fiber strength [43]. In this study, the hemicellulose content in *M. acuminata* and *M. balbisiana* is 5.76% and 5.60%, respectively. Lignin protects the plant against microorganism attack and conjointly influences the morphology, structural properties, bonding nature and wetness resistance of the fiber [44]. In this study, *M. acuminata* and *M. balbisiana* contain 0.73% and 1.08% lignin, respectively, in comparison to cellulose from hop stem (*Humulus lupulus*), which contains 44% cellulose, 13% hemicellulose, and 26% lignin [45].

## 4. Conclusions

Nanocellulose was successfully extracted from the pseudostem of the banana tree. The findings showed that lignin and hemicellulose removal can be achieved through pre-treatment, bleaching, and acid hydrolysis. The nano size can be obtained from these processes and the initial weight was reduced by 62%. The diameter size of the CNF was successfully determined as 80.31 ± 21.3 nm using TEM. Evidence of the degradation of non-cellulosic materials was found in the FTIR spectra. For *M. acuminata*, the disappearance of peak 1154 cm^−1^ was due to C-O-C vibration hemicellulose. Meanwhile, the peak 1238 cm^−1^ was found to be associated with the C-O-C stretching of the aryl-alkyl ether linkage in lignin. For *M. balbisiana*, a solid and broad band at 3413 cm^−1^ was attributed to the stretching vibration of hydroxyl groups (-OH). The XRD analysis showed that the crystallinity index of CNF for *M. acuminata* was 65.68%, whereas *M. balbisiana* was higher at 75.37%. TGA analysis showed that *M. balbisiana* pseudostem started degrading at a higher temperature (280 °C) compared to *M. acuminata* at 263 °C.

## Figures and Tables

**Figure 1 membranes-12-00451-f001:**
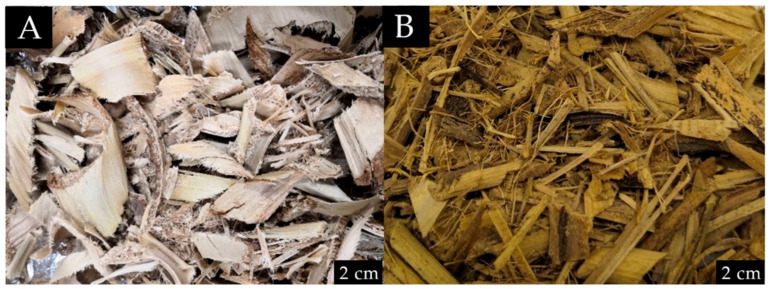
Difference between (**A**) dried *M. acuminata*; (**B**) dried *M. balbisiana* before grinding.

**Figure 2 membranes-12-00451-f002:**
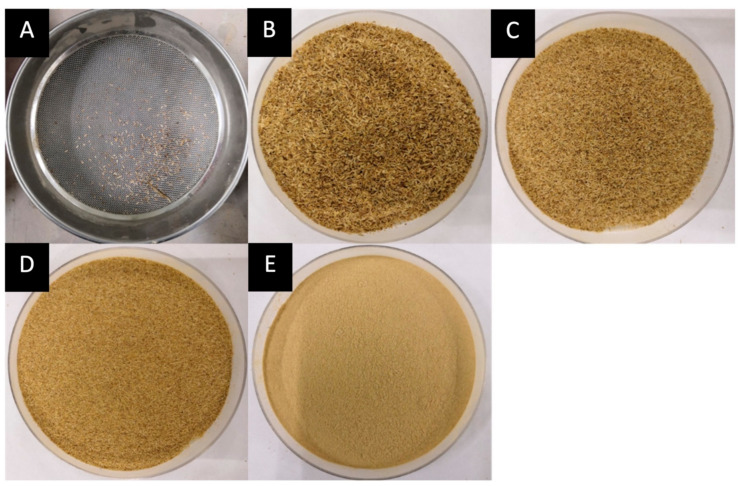
Fiber weight distribution according to mesh size (*M. acuminata*) of (**A**) <1 mm = 0.6474 g, (**B**) 1 mm–500 µm = 26.5757 g, (**C**) 500 µm–355 µm = 44.7802 g, (**D**) 355 µm–180 µm = 48.8895 g, (**E**) <180 µm = 114.6674 g.

**Figure 3 membranes-12-00451-f003:**
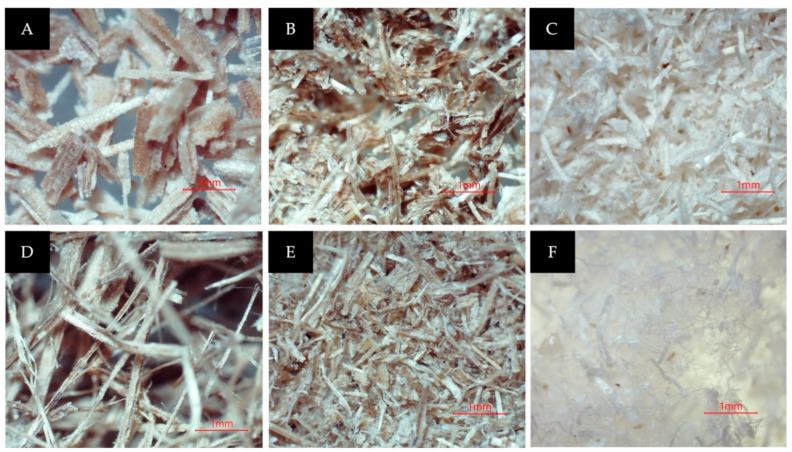
(**A**) *M. acuminata* pseudostem after being dried and sieved; (**B**) after the steam explosion; (**C**) after the bleaching process; (**D**) *M. balbisiana* pseudostem after being dried and sieved; (**E**) after the steam explosion; (**F**) after the bleaching process.

**Figure 4 membranes-12-00451-f004:**
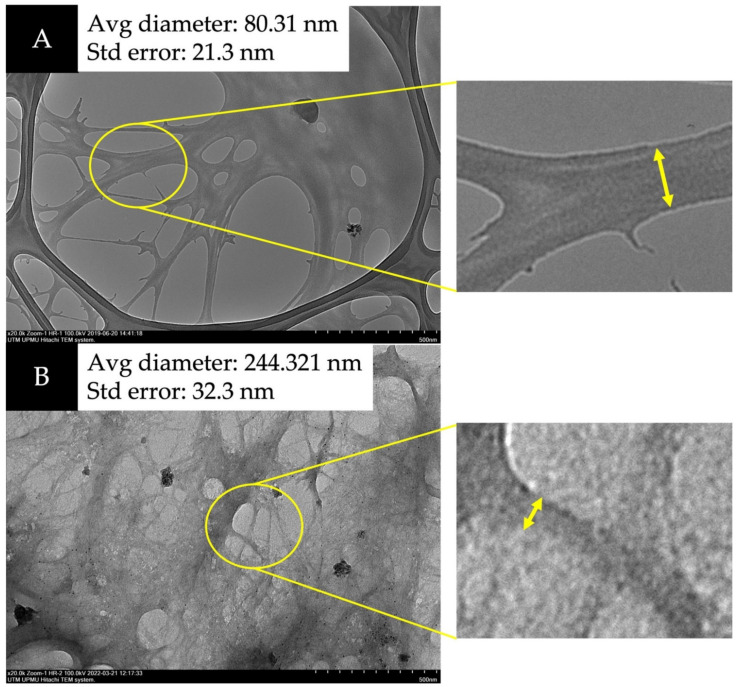
Transmission electron microscopy (TEM) images of (**A**) *M. acuminata* and (**B**) *M. balbisiana* at 20,000× magnification.

**Figure 5 membranes-12-00451-f005:**
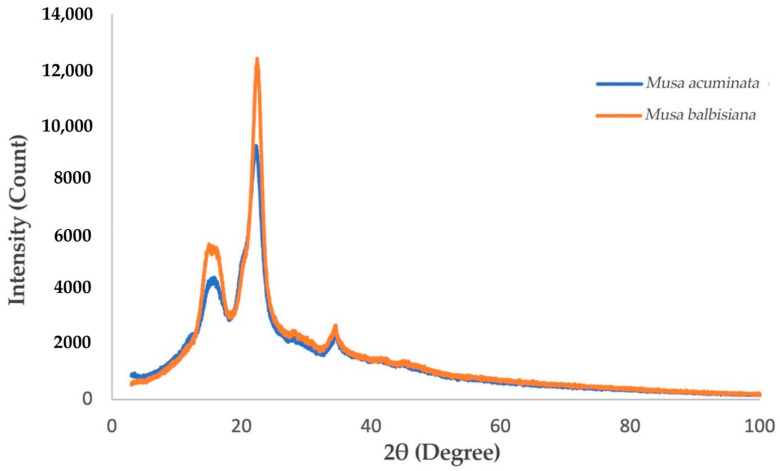
XRD Spectra of *M. acuminata* and *M. balbisiana* pseudostem CNF.

**Figure 6 membranes-12-00451-f006:**
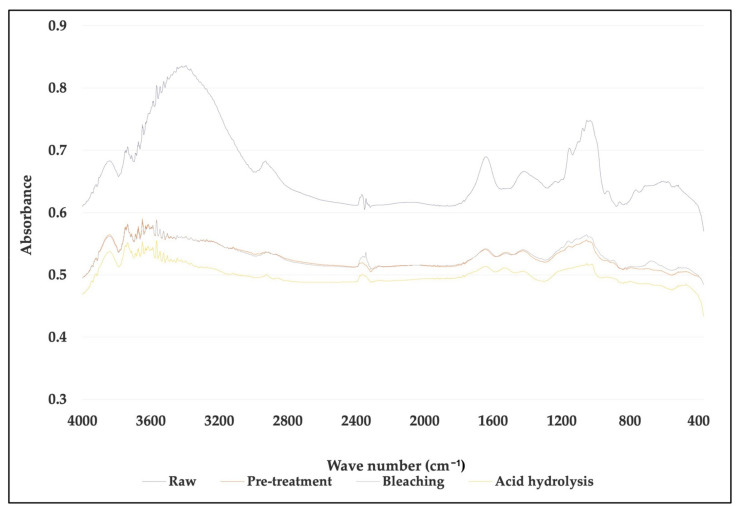
Comparison of FTIR spectra between untreated, treated fiber and cellulose nanofiber (CNF) for *Musa acuminata*.

**Figure 7 membranes-12-00451-f007:**
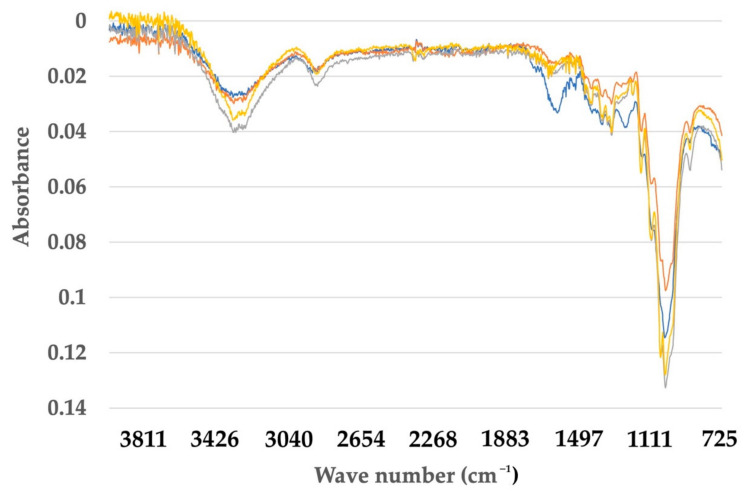
Comparison of FTIR spectra between untreated, treated fiber and cellulose nanofiber (CNF) for *Musa balbisiana*.

**Figure 8 membranes-12-00451-f008:**
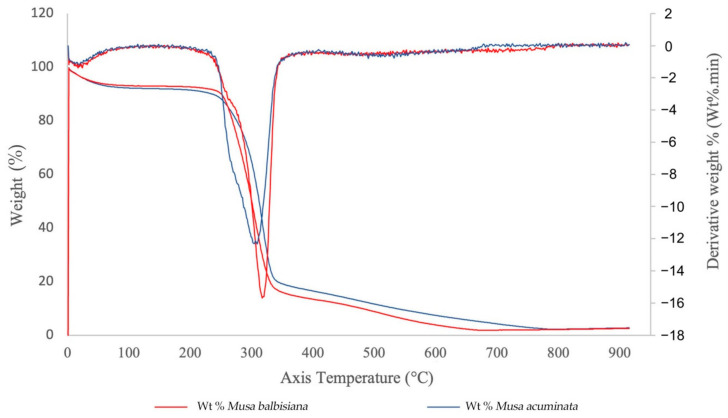
TGA curves of CNFs from *M. acuminata* and *M. balbisiana* pseudostem.

**Table 1 membranes-12-00451-t001:** Yield harvested after each process for *M. acuminata* and *M. balbisiana*.

Banana Pseudostem Sample	Process/Stage	Average Mass (%)
*Musa acuminata*	Initial Raw	100
	After Retting Process	64.35 ± 0.72
	After Steam Explosion Process	62.30 ± 1.52
	After Bleaching Process	46.70 ± 0.69
	After Acid Hydrolysis Process	40.50 ± 0.92
*Musa balbisiana*	Initial Raw	100
	After Retting Process	77.44 ± 0.64
	After Steam Explosion Process	37.58 ± 0.47
	After Bleaching Process	24.64 ± 1.07
	After Acid Hydrolysis Process	21.80 ± 0.57

**Table 2 membranes-12-00451-t002:** Lignocellulosic and extractives content of *M. acuminata* and *M. balbisiana* pseudostem before pre-treatment and after bleaching.

	Stages of Fiber Composition	Extractives (%)	Hemicellulose (%)	Lignin (%)	Cellulose (%)
*Musa acuminata*	Raw Fiber	1.80 ± 0.52	39.02 ± 1.45	26.58 ± 0.76	32.60 ± 1.79
	Fibre after pre-treatment and bleaching	0.35 ± 0.07	5.76 ± 0.46	0.73 ± 0.06	93.16 ± 0.50
*Musa balbisiana*	Raw Fiber	1.70 ± 0.08	42.26 ± 3.38	27.32 ± 2.48	28.72 ± 2.47
	Fiber after pre-treatment and bleaching	1.02 ± 0.10	5.60 ± 0.70	1.08 ± 0.06	93.36 ± 0.74

## Data Availability

Not applicable.

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
