# Peer review of "Preparation and Characterization of Cellulose Nanofibers from Banana Pseudostem by Acid Hydrolysis: Physico-Chemical and Thermal Properties"

_membranes, 2022, doi:10.3390/membranes12050451_

Round 1
Reviewer 1 Report
This paper described preparation and characterization of cellulose nanofibers from banana pseudostem by acid hydrolysis. The research results have reference value for its high-value utilization. Since the reported results were meaningful in this field, I recommend it to be published after minor revision.
1、The abstract of the manuscript is not refined enough to highlight the innovation. It may be better to let the advantages of the concept can be understood clearly.
2、The research status ofintroductionr part is relatively few, please supplemen.
3、The mechanism of the membrane formation is missing from the introduction section.
4、The scientific discussion and comparison for other research should be added.
Author Response
Please see the attachment. Thank you and have a nice ay.

Reviewer 2 Report
The work “Preparation and Characterization of Cellulose Nanofibers from Banana Pseudostem by Acid Hydrolysis: Physico-chemical and Thermal Properties” reports an approach to obtain NCs from the pseudo stem of banana tree after removing the lignin and hemicellulose through pretreatment, bleaching, and acid hydrolysis. The detailed comments are showed in below, major revisions are suggested for the publication in Membranes after significant improvements.
- For better understand the size of the materials, the scale plate should be added in Figure 1 and 3.
- Figure 4 only presents the TEM images of Musa acuminata CNF, the TEM images of Musa balbisiana CNF also should be characterized.
- The resolution ratio of Figure 5, 6 and 7 should be improved, and the annotations in Figure 6 are not clear, please redraw it.
- The degradation steps of M. acuminata and M. balbisiana are only undergoing one-step, some previous works have been reported the TG curves of CNF, such as:
1) DOI: 10.1016/j.matdes.2021.110379;
2) DOI: 10.1021/acssuschemeng0c02968;
3) DOI: 10.1016/j.carbpol.2021.119011
Please compare and analyze the reason of the difference.
Author Response
Please see the attachment. Thank you and have a nice day.

Reviewer 3 Report
The article titled " Preparation and Characterization of Cellulose Nanofibers from Banana Pseudostem by Acid Hydrolysis: Physico-chemical and Thermal Properties" describe in an acceptable way the characterization of cellulose nanofibers from this source.
However, there are things that must be improved for a best understanding of the readers and more specific.
Abstract: Line 19-20: " This research paper focuses on the production of banana pseudo stem cellulose nanofiber-based biofilm for packaging applications" when no application is done in this research.
Introduction: The use of CNF from banana pseudostem have been used in the past by other authors. I suggest to make a revision of the state of the art and include this information and explain the differences with the new article.
Results:
I miss the comparison of the CNF composition produce with banana pseudostem and other more common raw materials in the production of CNFs such as a comparision with softwood and hardwood pulps (Example of other materials: https://doi.org/10.1016/j.ijbiomac.2022.02.074)
The acid hydrolysis treatment is used commonly in the production of nanocrystals with very high crystalline index (>85%, normally). However, in this case not all the amorphous cellulose have been removed. What the authors think about that? Explain and compare in the manuscript
Tables:
Table 1: Maybe it is better to write the average mass in porcentage instead of grammes. That would facilitate the understanding of the readings.
Table 2: I do not understand, there are not units. Authors think the measure of 10 samples is enough to determine the size of the samples?
Table 3. Mistake: in extractives is written 1,7000 instead of 1,70
Conclusions:
Line 432-433: Authors refers to size o diameter of the samples?
Line 434. Authors refers to two peaks that dissapear but in the conclusion that is not relevant. The relevant part is the meaning of dissapearing this two peaks.
Author Response

(The authors gave the same response as above.)

Round 2
Reviewer 2 Report
The authors answered the comments in detail and improved their manuscript. I suggest this work can be published now.
Reviewer 3 Report
Comments and suggestions have been all changed.